# Politically Connected Independent Commissioners and Independent Directors on the Cost of Debt

**Onong Junus** [1,2] **, Iman Harymawan** [1,*] **, Mohammad Nasih** [1] **and Muslich Anshori** [1]

1   Department of Accountancy, Airlangga University, Surabaya 60286, Indonesia; onong.junus@gmail.com (O.J.);
    mohnasih@feb.unair.ac.id (M.N.); slich@feb.unair.ac.id (M.A.)
2   Faculty of Economics, Gorontalo University, Limboto 96211, Indonesia
*   Correspondence: harymawan.iman@feb.unair.ac.id

**Abstract:** This study examines the relationship between politically connected independent commissioners and independent directors regarding the cost of debt. The sample is all companies listed on the Indonesia Stock Exchange for the 2010–2017 period, totaling 327 companies with a total data value of 1722 firm-year observations. We used the ordinary least squares regression model (OLS) and the Heckman 2SLS method to solve the endogeneity problem. We found that politically connected independent commissioners and politically connected independent directors negatively correlate with the cost of debt. These results indicate the importance of politically connected independent commissioners and independent directors in managing companies, especially in obtaining loans with low interest rates. In addition, our results are robust due to the use of the Heckman 2SLS test. Therefore, this research can contribute to the development of the literature related to corporate governance and political connections in public companies, so that politically connected independent commissioners and independent directors have an essential role in decision-making in companies.

**Keywords:** independent commissioner; independent director; political connections; cost of debt

## 1. Introduction

Previous research has extensively studied the value of political connections within companies. The influence of political connections within a company can affect the company's value. This influence can improve company performance (Han and Zhang 2018), as well as increase the confidence of investors who want to invest (Maaloul et al. 2018) and have the opportunity to invest, and thus obtain a low cost of debt if the firm is politically connected (Chkir et al. 2020; Tee 2018).

Companies that tend to have political connections sometimes have the opportunity to obtain a low cost of debt compared to non-politically connected companies (Bliss et al. 2018). This tendency is similar to companies wherein the members of the board of directors have political ties and have the opportunity to obtain borrowing costs and lower ratings than lending banks (Houston et al. 2014) because politically connected companies have high ratings for paying off their debts, and the government guarantees to pay off the company's debts if the company is in poor financial condition (Tee 2018).

Politically connected companies have an increased chance of being selected by banks for loans because the political connections owned by the companies are good indicators for providing credit (Cheng and Wu 2019; Khaw et al. 2019), even if the company is experiencing problems in providing credit. In finance, banks still provide opportunities to grant loans to companies through political connections (Shi et al. 2020). This is the case in China and Indonesia, where politically connected companies have an excellent opportunity to access loans from commercial and state banks (Cheng and Wu 2019; Fu et al. 2017; Shi et al. 2020).

However, other evidence suggests that the role of a corporation's political connections reverses from what was described above. The research results from Bliss and Gul (2012) show that in Malaysia, politically connected companies have high leverage and high loan interest rates and are considered high risk because they tend to report high losses compared to nonpolitically connected companies. This is in line with the research conducted by Chaney et al. (2011) that found that politically connected companies experience high debt costs compared to non-politically connected companies. Thus, it can be concluded that there has not been a synchronization of research on political connections with the cost of debt, so more in-depth research is needed on the sources of political connections within companies.

In this study, we use a sample consisting of Indonesian listed companies. Indonesia is a developing country that is very interesting for the study of the relationship between business and politics because the political conditions in Indonesia changed massively after the collapse of the second former president Suharto in 1998. Furthermore, there was also a change in power from a centralized to a decentralized system (Joni et al. 2020). Political connections in Indonesia are standardisedin public companies by placing people who have close relations with the government in the company's organizational structure, both as commissioners and directors. This has happened from former president Susilo Bambang Yudhoyono's era to the current president Joko Widodo's administration by appointing commissioners of State-Owned Enterprises (BUMN) from political parties or volunteers to serve as commissioners. This happens not only to state companies but also to private companies. The facts prove that 23% of companies in Indonesia are politically connected (Chaney et al. 2011), increasing to 36% of companies being politically connected in a later study (Habib et al. 2017). In Indonesia, politically connected companies obtain loans from banks in two ways. First, companies can receive loans from state banks. Second, the loans obtained by companies exceed the company's loan application (Fu et al. 2017). This happens because creditors believe that loans given to politically connected companies can be repaid (Abiprayu 2021).

Indonesia adheres to a two-tier system of corporate governance, where the two boards consist of the Board of Commissioners and the Board of Directors. In OJK regulation No. 33/POJK.04/2014, at least one member of the Board of Commissioners and the Board of Directors comes from an independent party. Independent in this case means this member comes from outside of the company, has no affiliation with the major shareholders, has no relationship with the board of commissioners and directors, and does not have a relationship with the board of commissioners or directors of other companies. According to prior a study, a company is defined as politically connected if one of the principal shareholders or one of the leaders of the company (President, Vice President, CEO, Chairman, or secretary) is a minister, member of parliament, or is correlated with politicians or political parties (Faccio 2006). This demonstrates that the company's commissioners and directors can come from outside the company with a record of having political relations, because if one of the directors is an independent director who had previous political relations, they still make a positive contribution to the company (Ang et al. 2013).

This study focuses on independent commissioners and independent directors who have political connections; therefore, this research is crucial since companies with many independent directors tend to disclose reasonable and transparent financial statements (Chung and Zhang 2011; Tee 2019). Other evidence also explains that the political relationship owned by one of the independent directors is precious to minority shareholders, even though controlling shareholders often take over their interests (Hu et al. 2020). The value of a company's shares will decrease by 3.61% if the company loses an independent director from a government (Lei 2018), but other research reveals that politically connected companies pay higher debt costs than politically unconnected companies (Bliss and Gul 2012). From the previous explanation, the research question in this study is whether the political relationship between independent commissioners and independent directors is related to the cost of debt. The sample of this study uses companies listed on the Indonesia

Stock Exchange (IDX) from 2010 to 2017 and uses ordinary least squares (OLS) regression and the Heckman 2SLS method for analyzing and testing research hypotheses.

The results of this study show that both politically connected independent commissioners and politically connected independent directors have a negative significant relationship to the cost of debt. This means that independent commissioners and independent directors who are politically connected can reduce the cost of debt on corporate loans. We tested these results again using Heckman 2SLS, and the results were the same and confirmed the results of the previous OLS regression.

This research contributes to the development of the literature on corporate governance, political connections, and more specifically for public companies with independent commissioners and independent directors who have political connections in those companies.

This research is divided into five parts. The first part contains the background, the second part consists of the literature review and hypothesis development, the third part contains the research methods, the fourth part contains results and discussion, and the fifth part is the conclusion.

## 2. Literature Review

### 2.1. Prior Studies

The agency theory explains that the duties and responsibilities of company management increase the company's value, but managing the company cannot be separated from problems that occur both within the company and outside the company, which in turn creates problems between company shareholders and management. (Jensen and Meckling 1976). One of the functions of a good corporate governance system is the reduction of conflicts between management and shareholders. Companies that have many members on the board of commissioners will, of course, exercise their ability to monitor their business efforts, which can avoid agency conflicts and make it easier to obtain credit with low interest rates (Stefany and Joni 2020).

In the research of Ben-Nasr et al. (2021), companies that carry out board reform can reduce bank debt ratios because board reform and bank debts are substitutes for monitoring managers' actions. There are two sources of corporate debt financing: the first is from bonds from the public investors and the second is from direct loans to financial institutions, and owning one of them can increase the contribution to debt market development (Ben-Nasr et al. 2021). Various factors are considered when deciding the type of debt financing to take by the company, such as managerial ownership (Denis and Mihov 2003), the divergence of control-ownership from the controlling owner (Lin et al. 2013), multiple large shareholders (Boubaker et al. 2017), gender diversity, and board demographic factors (Cimini 2022).

To obtain low interest rates on loans provided by borrowers, a company must have power both at the level of the board of commissioners and the board of directors. That strength is a political connection (Keefe 2019). Since political connection is one of the conditions in conducting credit analysis that banks will provide as lenders (Houston et al. 2014), politically connected companies have higher debt repayment rates (Tee 2018). With political connections, banks can consider companies eligible for low interest loans (Liedong et al. 2015). Even banks as lenders assess that the political connections built by companies are capital owned by companies in improving relationships. Companies with lenders obtain loans with low interest rates (Houston et al. 2014), so having political connections within the company can lower the company's borrowing costs (Tee 2019).

Political connections can come from the board of commissioners, the board of directors, the company secretary, or the company's shareholders, but this research will only focus on the independent board of commissioners and directors. The existence of independent commissioners in the company can affect the company's cost of debt (Chen et al. 2009). Independent Commissioners and Independent Directors are managers tasked with contributing more to obtaining access to loans from outside the company because independent commissioners and independent directors are parties from outside the company who have no direct or indirect relationship with company owners or company shareholders

(Utomo et al. 2018). An independent commissioner who is also a representative of the company's minority shareholders and the audit committee is in charge of monitoring the directors' performance, while an independent director is in charge of monitoring the work of the executive board and minimizing conflicts of interest between the managers and company owners (Tanjung 2020).

*2.2. Hypothesis Development*

Independent commissioners are part of the implementation of good corporate governance in public companies and are also a manifestation of the independence and transparency of a company. If the company has a large proportion of independent commissioners, it can reduce its cost of debt (Anderson et al. 2004). The in detail disclosure of independent commissioners in the company is a form of company transparency to the public, and disclosing the company's human resources can reduce the company's debt costs (Putra et al. 2020). The low cost of debt is often exploited by companies with politically connected boards of commissioners rather than companies that are not politically connected (Joni et al. 2020). From the explanation above, the hypotheses in this study are as follows:

**Hypothesis 1 (H1).** *Politically connected independent commissioners have a negative relationship to the cost of debt.*

In addition to independent commissioners, independent directors also have a very important role in the sustainability of the company because the benefits of the presence of an independent director in the company will reduce the company's business risk (Chu et al. 2019). The research of Bradley and Chen (2015) found that the company's independent directors can reduce the cost of debt if credit conditions are strengthened, and with a higher percentage of busy independent directors, the company's cost of debt will decrease (Chakravarty and Rutherford 2017). Companies whose boards of directors are politically connected can reduce the cost of debt (Joni et al. 2020). Based on this, the second hypothesis of this study is:

**Hypothesis 2 (H2).** *Politically connected independent directors have a negative relationship to the cost of debt.*

**3. Methods**

*3.1. Sample Selection*

The sample of this research is companies listed on the Indonesia Stock Exchange (IDX) from 2010 to 2017, with a total unique firm being 327 firms. The number of samples is the final number after eliminating missing data on the variables used in this study. The year of the data observation is limited to 2017, because in 2018, IDX abolished the position of independent director at companies listed on the IDX through the Decree of the Directors of the Indonesia Stock Exchange Number: Kep-00183/BEI/12-2018. The research data source comes from the company's annual report and the ORBIS database, with 1722 firm-year observations. This study uses an ordinary least squares regression model (OLS), and we have conducted a winsorize on all variables at the 1 and 99 percentile levels to reduce the outliers.

In this study, there may be endogeneity problems in the research design; therefore, to overcome these problems, we conducted an endogeneity analysis using the Heckman 2SLS model as used by Harymawan and Nowland (2016).

The Heckman model has two regression stages. In the first stage, changing the position of the PCON_CI and PCON_DI variables, which were previously independent variables, into the dependent variable, then the instrumental variable (IV) and the control variables are added. The IV was obtained from the percentage of the political connections of independent commissioners and independent directors in companies based on industry classification standards (SIC). Since these are politically connected companies, both the

board of commissioners and the board of directors exist in each industry (Agrawal and Konoeber 2001; Harymawan and Nowland 2016). The instrument variables are named: PROBCON_CI and PROBCON_DI. The second stage of the Heckman 2SLS test retested H1 and H2 by entering the Inverse Mill's Ratio (IMR) value as an independent variable. IMR is the result of the first stage of 2SLS regression. IMR is used to explain whether or not there is a potential for self-selection bias in the regression; if the IMR coefficient value is not statistically significant, it means that there is no self-selection bias (Harymawan 2018). The IMRs are named: MIILS_CI and MIILS_DI.

*3.2. Variable Definition and Measurement*

3.2.1. Politically Connected Independent Commissioners and Independent Directors

Independent commissioners and independent directors who have political ties to this research are (1) those who come from former government agency officials; (2) law are those who come from a former judge or prosecutor; (3) those who are former politicians; (4) military are those who were formerly military or police; (5) political parties (Harymawan 2020; Shin et al. 2018). In this study, independent commissioners and directors related to politics were measured using a dummy variable, where the value is 1 if the company has independent commissioners and directors related to politics and 0 otherwise.

3.2.2. Cost of Debt

The cost of debt in this study is the dependent variable, which follows the same measurement as (Bliss et al. 2018), the cost of interest divided by the average total debt. Following several studies such as: (Bliss and Gul 2012; Khaw et al. 2019; Putra et al. 2020; Tee 2018), the cost of debt is measured by interest expense divided by the total average of long-term and medium-term debt, where interest expense is reported in the income statement, while the total long-term debt and short-term debt are reported in the balance sheet, and both reports are in the published annual report. The name and definition of the variable are shown in Table 1.

**Table 1.** Variable Definition and Measurement.

| Variable | Symbol | Measurement | Data Source |
|---|---|---|---|
| Independent Variable: | | | |
| Independent Commissioner connected to politics | PCON_CI | Dummy variable, a value of 1 if the independent commissioner has political connections and 0 otherwise. | Annual report |
| Independent directors connected to politics | PCON_DI | Dummy variable, a value of 1 if the company has independent directors connected to politics and 0 otherwise. | Annual report |
| Dependent variable: | | | |
| Cost of Debt | COD | The cost of debt is interest expense divided by total average debt. | OSIRIS |
| Control variable: | | | |
| Company Age | AGE | The length of time since the company was founded. | OSIRIS |
| Current Ratio | CRATIO | The current ratio is measured by its current assets divided by its current liabilities. | OSIRIS |
| Company Size | LN_TASSET | Natural logarithm of total assets | OSIRIS |
| Growth | ΔSALES | The sales value in year t minus the sales value in year t − 1 divided by the sales value in year t − 1 | OSIRIS |
| Number of commissioners | COMSIZE | Number of members of the company's board of commissioners | Annual report |
| Number of directors | DIRSIZE | Number of members of the company's board of directors | Annual report |
| Independent commissioner presentation | CI_PERCEN | Percentage of independent commissioners of the company | Annual report |
| Independent director presentation | DI_PERCEN | Company independent director presentation | Annual report |

Note: This table presents the proxy variables and variable definitions used in this study and measurements and data sources of the variables that were used.

## 4. Findings & Discussion

### 4.1. Sample Distribution of Politically Connected Independent Commissioners and Independent Directors

In this section, the distribution of the value of the number of independent commissioners and independent directors connected to politics is presented based on industry classification standards from 2010 to 2017, as presented in Table 2 below.

**Table 2.** Distribution of Politically Connected Independent Commissioners Based on Standard Industry Classification (SIC) 2010–2017.

| SIC | Commissioner Independent (CI) | | Total | % PCON_CI |
|---|---|---|---|---|
| | Connect | No Connect | | |
| (SIC 0) Agriculture, forestry and fisheries | 47 | 25 | 72 | 65.28 |
| (SIC 1) Mining | 159 | 63 | 222 | 71.62 |
| (SIC 2) Construction industries | 173 | 260 | 433 | 39.95 |
| (SIC 3) Manufacturing | 107 | 142 | 249 | 42.97 |
| (SIC 4) Transportation, communications and utilities | 210 | 105 | 315 | 66.67 |
| (SIC 5) Wholesale and retail trade | 71 | 55 | 126 | 56.35 |
| (SIC 6) Finance, Insurance and real estate | 100 | 41 | 141 | 70.92 |
| (SIC 7) Service industries | 71 | 69 | 140 | 50.71 |
| (SIC 8) Health, legal, and educational services and consulting | 18 | 6 | 24 | 75.00 |
| **Total** | **956** | **766** | **1722** | **55.52** |

Note: This table presents the distribution of politically connected independent commissioners based on SIC in 327 companies listed on IDX from 2010 to 2017.

Table 2 shows that during the 2010–2017 standard industry classification (SIC), there were 956 politically connected independent commissioners (CI) and 772 independent commissioners who did not have political connections, with the percentage of politically connected independent commissioners (PCON_CI) being 55.52%.

Table 3 shows that from the distribution of the data based on SIC, there are 98 politically connected independent directors (DI) and 1630 independent directors who do not have political connections, with a 5.69% percentage of politically connected independent directors (PCON_DI).

**Table 3.** Distribution of Politically Connected Independent Directors by Standard Industry Classification 2010–2017.

| SIC | Director Independent (DI) | | Total | % PCON_DI |
|---|---|---|---|---|
| | Connect | No Connect | | |
| (SIC 0) Agriculture, forestry and fisheries | 10 | 62 | 72 | 13.89 |
| (SIC 1) Mining | 26 | 196 | 222 | 11.71 |
| (SIC 2) Construction industries | 10 | 423 | 433 | 2.31 |
| (SIC 3) Manufacturing | 21 | 228 | 249 | 8.43 |
| (SIC 4) Transportation, communications and utilities | 24 | 291 | 315 | 7.62 |
| (SIC 5) Wholesale and retail trade | 2 | 124 | 126 | 1.59 |
| (SIC 6) Finance, Insurance and real estate | 3 | 138 | 141 | 2.13 |
| (SIC 7) Service industries | 2 | 138 | 140 | 1.43 |
| (SIC 8) Health, legal, and educational services and consulting | 0 | 24 | 24 | 0.00 |
| **Total** | **98** | **1624** | **1722** | **5.69** |

Note: This table presents the distribution of politically connected independent directors based on SIC in 327 companies listed on IDX from 2010 to 2017.

### 4.2. Descriptive Statistics

The results of the descriptive statistical tests of the research variables are presented in the following Table 4.

**Table 4.** Descriptive Statistics.

|  | Mean | Median | Standard Deviation | Minimum | Maximum |
|---|---|---|---|---|---|
| PCON_CI | 0.555 | 1.000 | 0.497 | 0.000 | 1.000 |
| PCON_DI | 0.057 | 0.000 | 0.232 | 0.000 | 1.000 |
| COD | 0.043 | 0.041 | 0.024 | 0.000 | 0.118 |
| AGE | 36.666 | 34.000 | 17.952 | 9.000 | 119.000 |
| CRATIO | 1.699 | 1.330 | 1.423 | 0.130 | 9.120 |
| LN_TASSET | 21.778 | 21.779 | 1.624 | 17.739 | 25.326 |
| ΔSALES | 0.085 | 0.072 | 0.322 | −0.748 | 1.695 |
| COMSIZE | 4.401 | 4.000 | 1.809 | 2.000 | 11.000 |
| DIRSIZE | 4.822 | 5.000 | 1.789 | 2.000 | 10.000 |
| CI_PERCEN | 37.201 | 33.333 | 13.937 | 0.000 | 75.000 |
| DI_PERCEN | 11.215 | 0.000 | 13.367 | 0.000 | 50.000 |

Note: This table shows the descriptive statistical results of the variables used in this study. This study took samples of all companies listed on the Indonesia Stock Exchange (IDX) for the 2010–2017 period, totaling 327 companies.

Table 4 above shows that the average value of politically connected independent commissioners (PCON_CI) is 0.555 with a standard deviation of 0.497, the value for politically connected independent directors (PCON_DI) is 0.057 with a standard deviation of 0.232, while the mean cost of debt (COD) is 0.043 with a standard deviation of 0.024, and the mean age of the companies (AGE) is 36,666 with the lowest company age at 9 years and the highest at 119 years. The mean value of the company's current ratio (CRATIO) is 1.699 with a standard deviation of 1.423; for company size (LN_TASSET) the mean value is 21.778 with a standard deviation of 1.624; the mean growth (ΔSALES) is 0.085, with the lowest growth value of −0.748 and the highest growth value of 1.695. The size of the board of commissioners (COMSIZE) has a mean value of 4401 with a standard deviation of 1809; while the minimum number of members on the board of commissioners is two people and a maximum of 11 people, the average number of boards of directors in a company (DIRSIZE) is 4.822, where the number of members on the board of directors is at least two people and at most ten people. The presentation of independent commissioners on the board of commissioners (CI_PERCEN) is 37.201 with a standard deviation of 13.937, the smallest presentation is 0.000, and the largest is 75.000. The presentation of independent directors on the board of directors (DI_PERCEN) was 11.215, with the smallest presentation of 0.000 and the largest of 50.000.

*4.3. Pearson Correlation*

In this section, the Pearson correlation results of the politically connected independent commissioners corresponding to the cost of debt and independent directors are politically connected to the cost of debt, as presented in Table 5.

Table 5 shows the results of the person correlation: statistically, there is a relationship between politically connected independent commissioners (PCON_CI) and the cost of debt (COD), while the politically connected independent director (PCON_DI) has no relationship to the cost of debt (COD), and the results of the relationship between control variables such as firm age (AGE), current ratio (CRATIO), and firm size (LN_ASSET) have a relationship with the COD with a significance level at 1%, but growth (ΔSALES) does not show a relationship to the COD. The number of commissioners (COMSIZE), the number of directors (DIRSIZE), and the presentation of independent commissioners (CI_PERCEN) have a significant relationship to COD, with a significance level of 1%.

**Table 5.** Pearson Correlation between Politically Connected Independent Commissioners and Independent Directors on The Cost of Debt.

| | COD | PCON_CI | PCON_DI | AGE | CRATIO | LN_TASSET | ΔSALES | COMSIZE | DIRSIZE | CI_PERCEN | DI_PERCEN |
|---|---|---|---|---|---|---|---|---|---|---|---|
| COD | 1.000 | | | | | | | | | | |
| PCON_CI | −0.090 *** | 1.000 | | | | | | | | | |
| | (0.000) | | | | | | | | | | |
| PCON_DI | −0.023 | 0.114 *** | 1.000 | | | | | | | | |
| | (0.347) | (0.000) | | | | | | | | | |
| AGE | −0.082 *** | 0.044 * | −0.084 *** | 1.000 | | | | | | | |
| | (0.001) | (0.067) | (0.000) | | | | | | | | |
| CRATIO | −0.084 *** | 0.033 | −0.059 ** | 0.007 | 1.000 | | | | | | |
| | (0.001) | (0.176) | (0.015) | (0.765) | | | | | | | |
| LN_TASSET | −0.087 *** | 0.387 *** | 0.028 | 0.166 *** | −0.007 | 1.000 | | | | | |
| | (0.000) | (0.000) | (0.238) | (0.000) | (0.783) | | | | | | |
| ΔSALES | −0.035 | 0.002 | −0.024 | −0.015 | 0.032 | 0.071 *** | 1.000 | | | | |
| | (0.150) | (0.931) | (0.311) | (0.541) | (0.190) | (0.003) | | | | | |
| COMSIZE | −0.188 *** | 0.348 *** | 0.019 | 0.102 *** | 0.021 | 0.533 *** | 0.018 | 1.000 | | | |
| | (0.000) | (0.000) | (0.430) | (0.000) | (0.385) | (0.000) | (0.447) | | | | |
| DIRSIZE | −0.189 *** | 0.221 *** | 0.027 | 0.094 *** | −0.011 | 0.563 *** | 0.012 | 0.504 *** | 1.000 | | |
| | (0.000) | (0.000) | (0.259) | (0.000) | (0.659) | (0.000) | (0.614) | (0.000) | | | |
| CI_PERCEN | 0.070 *** | 0.075 *** | 0.081 *** | 0.096 *** | 0.005 | 0.149 *** | 0.021 | 0.029 | 0.077 *** | 1.000 | |
| | (0.004) | (0.002) | (0.001) | (0.000) | (0.848) | (0.000) | (0.382) | (0.232) | (0.001) | | |
| DI_PERCEN | 0.126 *** | −0.030 | 0.145 *** | 0.023 | 0.004 | −0.017 | 0.020 | −0.126 *** | −0.178 *** | 0.177 *** | 1.000 |
| | (0.000) | (0.211) | (0.000) | (0.340) | (0.868) | (0.491) | (0.396) | (0.000) | (0.000) | (0.000) | |

Note: This table shows the results of the Pearson correlation of the variables used in this study. This study took samples of all companies listed on the Indonesia Stock Exchange (IDX) for the 2010–2017 period, totaling 327 companies. *p*-values in parentheses * $p < 0.1$, ** $p < 0.05$, *** $p < 0.01$.

*4.4. Independent t-Test*

In this section, the results of the independent *t*-test of the variables PCON_CI and PCON_DI are presented.

Table 6 presents the results of an independent *t*-test using 1722 samples to determine whether there is a difference between companies with independent commissioners connected to politics and independent commissioners who are not politically connected, where companies with independent commissioners that are politically connected totaling 956 companies and companies with independent commissioners not being politically connected amounting to 766. The results show that the average COD value for companies with politically connected independent commissioners is 0.041, while for companies with independent commissioners who are not politically connected the value is 0.046. The coefficient value shows that 0.004 is significant at the 1% level, and this indicates that there is a difference in the cost of debt (COD) between independent commissioners who are politically connected and independent commissioners who are not politically connected. The control variables in this study indicate that there is a difference, and there is no difference between companies with independent commissioners who have political relations with independent commissioners who do not have political relations, such as AGE, indicating that there are differences between companies with independent commissioners who have political relations with independent political ties to the commissioners, where the coefficient value is significant at the 10% level. On the other hand, the coefficient results from CRATIO show no difference between companies with independent commissioners who have political connections and independent commissioners who do not have political connections. The coefficient results from LN_TASSET showed a significant difference at the 1% level, while ΔSALES showed no difference. Furthermore, COMSIZE, DIRSIZE, and CI_PERCEN have a significant coefficient value at the 1% level, which means that there is a difference between independent commissioners who have political connections and independent commissioners who do not have political connections, and DI_PERCEN shows that there is no difference between independent commissioners who have political connections and independent commissioners who have no political connections.

**Table 6.** Independent *t*-test of Variable Politically Connected Independent Commissioner.

|  | PCON_CI | | Coef | *t*-Value |
|---|---|---|---|---|
|  | Mean Connect: 956 | Mean No_connect: 766 |  |  |
| COD | 0.041 | 0.046 | 0.004 *** | 3.743 |
| AGE | 37.376 | 35.779 | −1.596 * | −1.835 |
| CRATIO | 1.740 | 1.647 | −0.093 | −1.352 |
| LN_TASSET | 22.340 | 21.077 | −1.264 *** | −17.403 |
| ΔSALES | 0.086 | 0.085 | −0.001 | −0.087 |
| COMSIZE | 4.963 | 3.698 | −1.265 *** | −15.378 |
| DIRSIZE | 5.177 | 4.380 | −0.797 *** | −9.415 |
| CI_PERCEN | 38.131 | 36.041 | −2.090 *** | −3.101 |
| DI_PERCEN | 10.855 | 11.666 | 0.811 | 1.251 |

Note: the independent *t*-test above is based on 1722 observational data points from 327 companies listed on the Indonesia Stock Exchange (IDX), with a year of observation from 2010–2017. Significance at levels: * 10%, *** 1%.

Table 7 presents the results of an independent *t*-test, using 1722 samples to determine whether there is a difference between companies whose independent directors are politically connected and independent directors who are not politically connected, where there are 98 independent directors connected to politics and 1624 independent directors not connected to politics. The results show that the average COD value for companies with politically connected independent directors is 0.041, while companies with independent directors who are not politically connected have a value of 0.043. The coefficient value shows 0.002, which is not significant, and which means that there is no significant difference in the cost of debt (COD) between independent directors who are politically connected and

independent directors who are not politically connected. The control variables show that AGE and CRATIO show a difference between independent directors who are politically connected and independent directors who are not politically connected, where the coefficient values are significant at the 1% and 5% levels, respectively, while LN_TASSET, ΔSALES, COMSIZE, and DIRSIZE show no difference for independent directors that are politically connected and independent directors that are not politically connected. However, several control variables show significant differences at the 1% level, namely: CI_PERCEN and DI_PERCEN.

**Table 7.** Independent *t*-test of Variable Politically Connected Independent Director.

|  | PCON_DI | | Coef | *t*-Value |
|---|---|---|---|---|
|  | Mean Connect: 98 | Mean Connect: 1624 | | |
| COD | 0.041 | 0.043 | 0.002 | 0.940 |
| AGE | 30.500 | 37.038 | 6.538 *** | 3.512 |
| CRATIO | 1.358 | 1.719 | 0.361 ** | 2.442 |
| LN_TASSET | 21.966 | 21.767 | −0.199 | −1.181 |
| ΔSALES | 0.053 | 0.087 | 0.034 | 1.013 |
| COMSIZE | 4.541 | 4.392 | −0.149 | −0.790 |
| DIRSIZE | 5.020 | 4.810 | −0.210 | −1.129 |
| CI_PERCEN | 41.800 | 36.924 | −4.876 *** | −3.373 |
| DI_PERCEN | 19.080 | 10.741 | −8.339 *** | −6.060 |

Note: the independent *t*-test above is based on 1722 observational data points from 327 companies listed on the Indonesia Stock Exchange (IDX), with the year of observation from 2010–2017. Significance at levels: ** 5%, *** 1%.

### 4.5. OLS Regression Results

In this section, the results of OLS regression will be explained as a proof of the hypothesis presented in this study.

Table 8 shows the OLS regression results of politically connected independent commissioners (PCON_CI) and politically connected independent directors (PCON_DI) corresponding to the cost of debt (COD). The results in model 1 show that PCON_CI has a negative and significant relationship to COD, where the coefficient value is −0.003 and the t-count value is −2.67 and is significant at the 1% level. The coefficient of determination (Adj R-squared) shows that the cost of debt (COD) can be explained by an independent commissioner with political connections (PCON_CI) of 10.7%. While model 2 in Table 8 explains a negative relationship between PCON_DI and COD with a coefficient value of −0.006 and t-count value of −2.37, respectively, statistically, the relationship is significant at the 5% level. Then the value of the coefficient of determination (Adj R-squared) shows that the cost of debt (COD) can be explained by a politically connected independent director (PCON_DI) value of 10.6%. In model 3, the researchers tried to perform a regression test simultaneously between PCON_CI and PCON_DI on COD, and the results were also the same as the results of the previous separate regression test.

**Table 8.** OLS Regression Results.

|  | (1) | (2) | (3) |
|---|---|---|---|
|  | COD | COD | COD |
| PCON_CI | −0.003 *** |  | −0.003 ** |
|  | (−2.67) |  | (−2.42) |
| PCON_DI |  | −0.006 ** | −0.005 ** |
|  |  | (−2.37) | (−2.09) |
| AGE | −0.000 *** | −0.000 *** | −0.000 *** |
|  | (−2.92) | (−3.05) | (−3.10) |
| CRATIO | −0.001 *** | −0.001 *** | −0.001 *** |
|  | (−2.62) | (−2.82) | (−2.74) |
| LN_TASSET | 0.001 ** | 0.001 | 0.001 ** |
|  | (2.14) | (1.52) | (2.03) |
| ∆SALES | −0.002 | −0.002 | −0.002 |
|  | (−1.07) | (−1.03) | (−1.11) |
| COMSIZE | −0.002 *** | −0.002 *** | −0.002 *** |
|  | (−3.84) | (−4.39) | (−3.85) |
| DIRSIZE | −0.002 *** | −0.002 *** | −0.002 *** |
|  | (−4.67) | (−4.47) | (−4.57) |
| CI_PERCEN | 0.000 ** | 0.000 *** | 0.000 *** |
|  | (2.54) | (2.60) | (2.65) |
| DI_PERCEN | 0.000 | 0.000 * | 0.000 * |
|  | (1.48) | (1.78) | (1.76) |
| Industry | Included | Included | Included |
| Year | Included | Included | Included |
| _cons | 0.039 *** | 0.044 *** | 0.040 *** |
|  | (3.92) | (4.59) | (4.05) |
| r2 | 0.119 | 0.119 | 0.122 |
| Adj R-squared | 0.107 | 0.106 | 0.109 |
| N | 1722 | 1722 | 1722 |
| Mean VIF | 2.31 | 2.30 | 2.27 |
| F | 9.59 | 9.52 | 9.40 |
| Prob > F | 0.0000 | 0.0000 | 0.0000 |
| f | 412.2704 | 89.5999 |  |
| 2 * Pr(F > f) | 0.0000 | 0.0000 |  |

Note: This table shows the OLS regression results of politically connected independent commissioners and politically connected independent directors on the cost of debt. This study took samples of all companies listed on the Indonesia Stock Exchange (IDX) for the period from 2010 to 2017, totaling 327 companies. t statistics in parentheses * $p < 0.1$, ** $p < 0.05$, *** $p < 0.01$.

The results of model 1 show that the politically connected independent commissioner (PCON_CI) value has a negative and significant relationship to the cost of debt (COD). This means that the presence of a politically connected independent commissioner on the board of commissioners can affect the company's low cost of debt; this is because of the trust and good communication built by the company's independent commissioners towards creditors, as well as the fact that state banks are more interested in companies that have political connections at the board of commissioners level and can thus be given loan funds (Abiprayu 2021; Fu et al. 2017). Furthermore, when viewed from the presentation of the company's independent commissioners, an average of 37.2%, the political connection of independent parties within the company's top management follows Faccio (2006), where the element of political connection in the company is at least 30%. The results of this study are consistent with Putra et al. (2020), whose work explains that the company's cost of debt will decrease if there is an independent commissioner in the company. The results of this study strengthen the research of Joni et al. (2020), which explains that companies with politically connected supervisory boards can reduce the cost of debt. This explanation verifies H1 in this study.

Model 2 from Table 8 shows that politically connected independent directors (PCON_DI) have a significant negative relationship to the cost of debt (COD). This means that politically

connected independent directors can reduce the company's cost of debt. The coefficient value of PCON_DI demonstrates this for the COD which has a significant negative value (−0.006) at the 1% level. So, the more independent directors connected to politics in the company, the better they are at managing finances, especially in obtaining loans with low interest rates. Therefore, companies need to present independent directors to reduce business risk (Chu et al. 2019). When credit conditions are strengthened, independent directors reduce the cost of debt (Bradley and Chen 2015). The results of this study are in line with the research of Houston et al. (2014) and Joni et al. (2020), where companies whose boards of directors are politically connected can reduce the cost of debt compared to companies whose directors are not politically connected. Based on this explanation, H2 in this study is verified.

Next, in the third model, the results of the joint regression of PCON_CI and PCON_DI are presented. The results found are the same as the statistical results in models 1 and 2, both of which show a negative and significant relationship between politically connected independent commissioners and politically connected independent directors regarding the cost of debt. Overall, the correlation results from the three models in Table 8 do not indicate a multicollinearity problem because the average value of VIF is below 10 (Belsley et al. 2005).

*4.6. Robustness Test*

There is a potential for an endogeneity problem in this research, where the variables contain elements of the selection of a policy; therefore, there is a variable that is not observed to be correlated with independent commissioners who have political connections, independent directors who have political connections correlated, and company performance. To overcome this problem, we use Heckman's 2SLS model as in previous research by (Kim and Zhang 2016).

At this stage in the first-stage regression, we look for the factors of the company connected with politics, as in the equation below

$$PCON\_CI = \beta1 + \beta2Control_{it} + \beta3Y_{it} + IndustryFE + YearFE + \varepsilon \tag{1}$$

$$PCON\_DI = \beta1 + \beta2Control_{it} + \beta3Y_{it} + IndustryFE + YearFE + \varepsilon \tag{2}$$

PCON_CI and PCON_DI are dummy variables as proxies for the political connections of independent commissioners and independent directors, Control_{it} is the control variable of this first regression, and Y_{it} is an additional variable where the determinants of independent commissioners and independent directors are connected to politics but have no effect on the cost of debt. To determine this, we follow research from Harymawan and Nowland (2016) and Kim and Zhang (2016) where the political connections of independent commissioners and independent directors of companies are grouped by company industry presentation (PROBCON_CI and PROBCON_DI). The regression results of the two equations above can be seen in Tables 9 and 10.

Table 9 is the first-stage regression from equation 1, where we include PROBCON_CI as an instrumental variable. As a result, we found that PROBCON_CI has a positive and significant effect on the politically connected independent commissioner (PCON_CI) value, where the coefficient value of 0.035 (t = 3.94) is significant at the 1% level. The control variables in this study showed that AGE and CRATIO did not affect PCON_CI LN_TASSET, COMSIZE, and DIRSIZE had a positive effect on PCON_CI, significant at the 1% and 10% levels. However, ΔSALES, CI_PERCEN, and DI_PERSEN do not affect PCON_CI. The significant results are shown by the instrument variable (PROBCON_CI) against PCON_CI, which means that the instrument variables used in the regression can be used to see which factors can affect the political relationship of the company's independent commissioners.

**Table 9.** Fist-Stage Regression Model Results: Determinants of Firm Political Connections From PCON_CI.

|  | (1) |
|---|---|
|  | **PCON_CI** |
| PCON_CI |  |
| PROBCON_CI | 0.035 *** |
|  | (3.94) |
| AGE | −0.002 |
|  | (−1.15) |
| CRATIO | 0.029 |
|  | (1.24) |
| LN_TASSET | 0.262 *** |
|  | (8.73) |
| ΔSALES | −0.150 |
|  | (−1.32) |
| COMSIZE | 0.210 *** |
|  | (7.91) |
| DIRSIZE | −0.048 * |
|  | (−1.93) |
| CI_PERCEN | 0.002 |
|  | (0.95) |
| DI_PERCEN | 0.001 |
|  | (0.47) |
| _cons | −8.275 *** |
|  | (−9.49) |
| Year FE | Yes |
| Industry FE | Yes |
| r2_p | 0.194 |
| N | 1722 |

Note: This table shows the results of the Heckman 2SLS first-stage regression model for the political connection determinant of PCON_CI. This study took samples of all companies listed on the Indonesia Stock Exchange (IDX) for the 2010–2017 period, totaling 327 companies. t statistics in parentheses * $p < 0.1$, *** $p < 0.01$.

Table 10 shows the first-stage regression results from Equation (2), where we include PROBCON_DI as the instrumental variable. As a result, we found that PROBCON_DI has a positive and significant effect on the politically connected independent directors (PCON_DI), where the coefficient value of 0.093 (t = 4.03) is significant at the 1% level. The control variables used in this study showed that AGE and CRATIO had a negative effect on PCON_DI, with significant levels at the 1% and 5% levels, but LN_TASSET, ΔSALES had no effect on PCON_DI. Furthermore, the variables COMSIZE, DIRSIZE, CI_PERCEN, and DI_PERCEN positively affect PCON_DI, with significant levels at 5% and 1%, respectively. The significance of the instrument variable (PROBCON_DI) against PCON_DI, means that the instrument variable used in this regression can be used to see which factors can influence the political connections of the company's independent directors.

The second-stage regression: in this section, a re-examination of H1 and H2 is carried out by entering the value of the inverse Mill's ratio of politically connected independent commissioners and politically connected independent directors. The regression equation for the second stage is as follows:

$$COD = \beta1 + \beta2\,PCON\_CI + \beta3 Control_{it} + \beta4 MILLS\_CI_{it} + IndustryFE + YearFE + \varepsilon \quad (3)$$

$$COD = \beta1 + \beta2\,PCON\_DI + \beta3 Control_{it} + \beta4 MILLS\_DI_{it} + IndustryFE + YearFE + \varepsilon \quad (4)$$

The results of the Second-stage regression can be seen in Table 11.

**Table 10.** Results of The Fist-Stage Regression Model: Determinants of Firm Political Connections From PCON_DI.

|  | (1) |
| --- | --- |
|  | **PCON_DI** |
| PCON_DI |  |
| PROBCON_DI | 0.093 *** |
|  | (4.03) |
| AGE | −0.019 *** |
|  | (−3.90) |
| CRATIO | −0.101 ** |
|  | (−2.19) |
| LN_TASSET | −0.056 |
|  | (−1.12) |
| ΔSALES | −0.067 |
|  | (−0.40) |
| COMSIZE | 0.070 ** |
|  | (2.16) |
| DIRSIZE | 0.105 *** |
|  | (2.69) |
| CI_PERCEN | 0.011 ** |
|  | (2.56) |
| DI_PERCEN | 0.031 *** |
|  | (5.56) |
| _cons | −1.685 |
|  | (−1.61) |
| Year FE | Yes |
| Industry FE | Yes |
| r2_p | 0.211 |
| N | 1722 |

Note: This table shows the results of the Heckman 2SLS first-stage regression model for the political connection determinant of PCON_DI. This study took a sample of all companies listed on the Indonesia Stock Exchange (IDX) from 2010 to 2017, amounting to 327 companies. t statistics in parentheses ** $p < 0.05$, *** $p < 0.01$.

Table 11 shows that the first model is a retest of the first hypothesis (Equation (3)). By including the inverse Mill's ratio variables (MILLS_CI and MILLS_DI) obtained in the first 2SLS regression, it was found that politically connected independent commissioners have a negative relationship with the cost of debt with the coefficient value (−0.004) and the t-count value (−2.65) and significant at the 1% level. This shows that the results of the Heckman 2SLS regression are the same as the OLS regression results that test H1, and that the presence of politically connected independent commissioners can reduce the company's cost of debt. Furthermore, the second model in Table 11 provides an overview of the results of the retest of the second hypothesis (Equation (4)); it was found that independent directors who are politically connected have a negative relationship to the cost of debt, where the coefficient value (−0.006) and t value (−2.62) and significant at the 1% level. This indicates that the results of the Heckman 2SLS regression are the same as the results of the previous OLS regression that tested H2. The presence of an independent director connected to politics in the company can reduce the cost of debt.

In this study, the instrument variables used were PROBCON_CI and PROBCON_DI, resulting in an insignificant inverse Mill's ratio (MILLS_CI and MILLS_DI). Not significant MILLS_CI and MILLS_DI mean that this instrumental variable is good at explaining the relationship between politically connected independent commissioners and politically connected independent directors with regard to the cost of debt.

**Table 11.** Results of Second-Stage Regression, Heckman 2SLS.

|  | (1) | (2) | (3) |
|---|---|---|---|
|  | COD | COD | COD |
| PCON_CI | −0.004 *** |  | −0.003 ** |
|  | (−2.65) |  | (−2.43) |
| PCON_DI |  | −0.006 *** | −0.005 ** |
|  |  | (−2.62) | (−2.28) |
| AGE | −0.000 *** | −0.000 *** | −0.000 *** |
|  | (−2.89) | (−2.94) | (−2.86) |
| CRATIO | −0.001 ** | −0.001 ** | −0.001 ** |
|  | (−2.09) | (−2.34) | (−2.39) |
| LN_TASSET | 0.001 | 0.001 | 0.000 |
|  | (0.60) | (1.09) | (0.38) |
| ΔSALES | −0.002 | −0.002 | −0.002 |
|  | (−0.70) | (−0.93) | (−0.85) |
| COMSIZE | −0.002 *** | −0.002 *** | −0.002 ** |
|  | (−2.83) | (−3.72) | (−2.48) |
| DIRSIZE | −0.002 *** | −0.002 *** | −0.002 *** |
|  | (−4.65) | (−3.48) | (−3.38) |
| CI_PERCEN | 0.000 ** | 0.000 *** | 0.000 *** |
|  | (2.32) | (2.81) | (2.74) |
| DI_PERCEN | 0.000 | 0.000 ** | 0.000 ** |
|  | (1.33) | (1.98) | (1.96) |
| MILLS_CI | −0.003 |  | −0.003 |
|  | (−0.64) |  | (−0.66) |
| MILLS_DI |  | 0.003 | 0.003 |
|  |  | (1.19) | (1.23) |
| _cons | 0.052 ** | 0.040 *** | 0.049 ** |
|  | (2.15) | (3.45) | (1.99) |
| Year FE | Yes | Yes | Yes |
| Industry FE | Yes | Yes | Yes |
| r2_p |  |  |  |
| r2_a | 0.107 | 0.106 | 0.109 |
| N | 1722 | 1722 | 1722 |
| Mean VIF | 3.30 | 3.59 | 4.38 |

Note: This table shows the results of the second-stage Heckman 2SLS regression. This study took samples from all companies listed on the Indonesia Stock Exchange (IDX) for the 2010–2017 period, totaling 327 companies. t statistics in parentheses ** $p < 0.05$, *** $p < 0.01$.

### 4.7. Additional Test

In an additional test in this study, we wanted to examine further analysis to the political connection and the cost of debt based on five categories of political connections (government agent officials, former law, former politicians, former military, and political parties). The results can be seen in the table below.

Based on Table 12 above, there are five categories of independent commissioners' political connections, namely (1) former government agency officials (PCON_GOVEROFF_CI), (2) former judges or prosecutors (PCON_LAW_CI), (3) former politicians (PCON_FORMERPOL_CI), (4) former military (PCON_MILITARY_CI), and (5) political parties (PCON_POLPARTIES_CI). Based on the regression results, it was found that the political connection category of independent commissioners with military backgrounds had a negative and significant relationship with the cost of debt. This is evidenced by the coefficient value, which shows a negative value (−0.003), and the t-count value (−2.12) which is significant at the 5% level. Meanwhile, the four categories of political connections: PCON_GOVEROFF_CI, PCON_LAW_CI, PCON_FORMERPOL_CI, and PCON_POLPARTIES_CI, had no relationship to the cost of debt.

**Table 12.** Regression of 5 categories of political connections independent commissioners to the cost of debt.

|  | (1) | (2) | (3) | (4) | (5) |
|---|---|---|---|---|---|
|  | **COD** | **COD** | **COD** | **COD** | **COD** |
| PCON_MILITARY_CI | −0.003 ** | | | | |
|  | (−2.12) | | | | |
| PCON_GOVEROFF_CI | | −0.002 | | | |
|  | | (−1.46) | | | |
| PCON_LAW_CI | | | −0.002 | | |
|  | | | (−0.32) | | |
| PCON_FORMERPOL_CI | | | | −0.002 | |
|  | | | | (−0.74) | |
| PCON_POLPARTIES_CI | | | | | 0.002 |
|  | | | | | (0.39) |
| AGE | −0.000 *** | −0.000 *** | −0.000 *** | −0.000 *** | −0.000 *** |
|  | (−2.84) | (−2.84) | (−2.84) | (−2.86) | (−2.83) |
| CRATIO | −0.001 *** | −0.001 *** | −0.001 *** | −0.001 *** | −0.001 *** |
|  | (−2.71) | (−2.66) | (−2.67) | (−2.63) | (−2.66) |
| LN_TASSET | 0.001 * | 0.001 * | 0.001 | 0.001 | 0.001 |
|  | (1.78) | (1.79) | (1.59) | (1.59) | (1.57) |
| ΔSALES | −0.002 | −0.002 | −0.002 | −0.002 | −0.002 |
|  | (−1.03) | (−1.03) | (−0.98) | (−0.96) | (−1.00) |
| COMSIZE | −0.002 *** | −0.002 *** | −0.002 *** | −0.002 *** | −0.002 *** |
|  | (−4.32) | (−4.07) | (−4.41) | (−4.29) | (−4.43) |
| DIRSIZE | −0.002 *** | −0.002 *** | −0.002 *** | −0.002 *** | −0.002 *** |
|  | (−4.55) | (−4.58) | (−4.58) | (−4.56) | (−4.59) |
| CI_PERCEN | 0.000 ** | 0.000 ** | 0.000 ** | 0.000 ** | 0.000 ** |
|  | (2.45) | (2.55) | (2.47) | (2.51) | (2.41) |
| DI_PERCEN | 0.000 | 0.000 | 0.000 | 0.000 | 0.000 |
|  | (1.43) | (1.48) | (1.43) | (1.48) | (1.45) |
| Industry | Included | Included | Included | Included | Included |
| Year | Included | Included | Included | Included | Included |
| _cons | 0.041 *** | 0.041 *** | 0.043 *** | 0.043 *** | 0.044 *** |
|  | (4.25) | (4.25) | (4.46) | (4.46) | (4.49) |
| r2 | 0.118 | 0.117 | 0.116 | 0.116 | 0.116 |
| Adj R-squared | 0.106 | 0.104 | 0.103 | 0.103 | 0.103 |
| *N* | 1722 | 1722 | 1722 | 1722 | 1722 |
| *Mean VIF* | 2.30 | 2.31 | 2.29 | 2.30 | 2.30 |

Note: This table shows the regression results of 5 categories of independent commissioners' political connections to the cost of debt. This study took samples of all companies listed on the Indonesia Stock Exchange (IDX) for the 2010–2017 period, totaling 327 companies. t statistics in parentheses * $p < 0.1$, ** $p < 0.05$, *** $p < 0.01$.

Hence, it is clear that the significance of the independent commissioner's relationship to politics in the OLS regression table model 8 (1) is formed from the category of the independent commissioner's political relationship with a military background. This result is in line with Harymawan (2018) who shows that companies that recruit personnel from the former military can reduce company loan interest costs.

Based on Table 13 above, the regression results from the five categories of independent directors' political connections are 1. Former politician (PCON_FORMERPOL_DI), 2. Former military (PCON_MILITARY_DI), 3. Former government agency official (PCON_GOVEROFF_CI), 4. Former judge or prosecutor (PCON_LAW_DI)), and 5. Political parties (PCON_POLPARTIES_DI). The results indicate that the political relationship category of independent directors with former politicians (PCON_FORMERPOL_DI) and former military members (PCON_MILITARY_DI) has a negative and significant relationship to the cost of debt. Where PCON_FORMERPOL_DI has a negative coefficient value (−0.028) which is significant at the 1% level, with t-count (−3.58), then PCON_MILITARY_DI has a negative coefficient value (−0.013) which is significant at the 5% level, with t-count (−2.13). Mean-

while, three categories of political connections, PCON_GOVEROFF_DI, PCON_LAW_DI, and PCON_POLPARTIES_DI, did not show a relationship with the cost of debt (COD).

**Table 13.** Regression of 5 types of independent directors' political connection to the cost of debt.

| | (1) | (2) | (3) | (4) | (5) |
|---|---|---|---|---|---|
| | **COD** | **COD** | **COD** | **COD** | **COD** |
| PCON_FORMERPOL_DI | −0.028 *** | | | | |
| | (−3.58) | | | | |
| PCON_MILITARY_DI | | −0.013 ** | | | |
| | | (−2.13) | | | |
| PCON_GOVEROFF_DI | | | −0.002 | | |
| | | | (−0.87) | | |
| PCON_LAW_DI | | | | 0.000 | |
| | | | | (0.01) | |
| PCON_POLPARTIES_DI | | | | | 0.000 |
| | | | | | (.) |
| AGE | −0.000 *** | −0.000 *** | −0.000 *** | −0.000 *** | −0.000 *** |
| | (−2.93) | (−2.89) | (−2.90) | (−2.83) | (−2.83) |
| CRATIO | −0.001 *** | −0.001 *** | −0.001 *** | −0.001 *** | −0.001 *** |
| | (−2.72) | (−2.73) | (−2.72) | (−2.69) | (−2.69) |
| LN_TASSET | 0.001 | 0.001 | 0.001 | 0.001 | 0.001 |
| | (1.18) | (1.39) | (1.59) | (1.58) | (1.58) |
| ΔSALES | −0.002 | −0.002 | −0.002 | −0.002 | −0.002 |
| | (−1.02) | (−0.99) | (−1.00) | (−0.98) | (−0.98) |
| COMSIZE | −0.002 *** | −0.002 *** | −0.002 *** | −0.002 *** | −0.002 *** |
| | (−4.29) | (−4.42) | (−4.40) | (−4.43) | (−4.43) |
| DIRSIZE | −0.002 *** | −0.002 *** | −0.002 *** | −0.002 *** | −0.002 *** |
| | (−4.59) | (−4.52) | (−4.52) | (−4.58) | (−4.58) |
| CI_PERCEN | 0.000 *** | 0.000 ** | 0.000 ** | 0.000 ** | 0.000 ** |
| | (2.64) | (2.53) | (2.50) | (2.47) | (2.47) |
| DI_PERCEN | 0.000 | 0.000 | 0.000 | 0.000 | 0.000 |
| | (1.54) | (1.63) | (1.55) | (1.45) | (1.45) |
| Industry | Included | Included | Included | Included | Included |
| Year | Included | Included | Included | Included | Included |
| _cons | 0.047 *** | 0.046 *** | 0.043 *** | 0.043 *** | 0.043 *** |
| | (4.83) | (4.71) | (4.48) | (4.48) | (4.48) |
| r2 | 0.122 | 0.118 | 0.116 | 0.116 | 0.116 |
| Adj R-squared | 0.111 | 0.106 | 0.104 | 0.103 | 0.104 |
| N | 1722 | 1722 | 1722 | 1722 | 1722 |
| *Mean VIF* | 2.30 | 2.31 | 2.30 | 2.29 | 2.35 |

Note: This table shows the regression results of 5 categories of independent directors' political connection to the cost of debt. This study took samples of all companies listed on the Indonesia Stock Exchange (IDX) for the 2010–2017 period, totaling 327 companies. t statistics in parentheses ** $p < 0.05$, *** $p < 0.01$.

These results indicate that the significance of the politically connected independent director relationship in the OLS regression Table 8 model (2) is formed from the category of the independent director's political connection with a background of former political members (PCON_FORMERPOL_DI) and former military members (PCON_MILITARY_DI). These results align with Bliss et al. (2018), who showed that corporate loan interest rates in China will decrease if the company directors come from the legislative assembly and members of the national people's congress come from the military (Harymawan 2018).

Finally, we examine the relationship between the politically connected independent commissioner (PCON_CI) and politically connected independent director (PCON_DI) and becoming PCON_CIDI, where the regression results are shown in Table 14.

**Table 14.** Regression of PCON_CIDI, PCON_CI, and PCON_DI on the cost of debt (COD).

|  | **(1)** |
|---|---|
|  | **COD** |
| PCON_CIDI | 0.010 * |
|  | (1.67) |
| PCON_CI | −0.003 *** |
|  | (−2.68) |
| PCON_DI | −0.013 ** |
|  | (−2.47) |
| AGE | −0.000 *** |
|  | (−3.11) |
| CRATIO | −0.001 *** |
|  | (−2.70) |
| LN_TASSET | 0.001 ** |
|  | (2.01) |
| ΔSALES | −0.002 |
|  | (−1.10) |
| COMSIZE | −0.002 *** |
|  | (−3.84) |
| DIRSIZE | −0.002 *** |
|  | (−4.58) |
| CI_PERCEN | 0.000 *** |
|  | (2.67) |
| DI_PERCEN | 0.000 * |
|  | (1.80) |
| Industry | Included |
| Year | Included |
| _cons | 0.041 *** |
|  | (4.14) |
| r2 | 0.123 |
| Adj R-squared | 0.110 |
| *N* | 1722 |
| *Mean VIF* | 2.52 |

Note: This table shows the regression results of PCON_CIDI, PCON_CI, and PCON_DI to the cost of debt. This study took samples of all companies listed on the Indonesia Stock Exchange (IDX) for the 2010–2017 period, totaling 327 companies. t statistics in parentheses * $p < 0.1$, ** $p < 0.05$, *** $p < 0.01$.

From the results of Table 14 above, it is evident that PCON_CIDI has a positive and significant relationship to COD, where the coefficient value (0.010) is significant at the 10% level with a t-count of (1.67). At the same time, PCON_CI and PCON_DI show both a negative and significant relationship to COD. Where PCON_CI has a negative coefficient (−0.003) and t count (−2.68) is significant at the 1% level, then PCON_DI has a negative coefficient value (−0.013), and t count (−2.47) is significant at the 5% level. Overall, the results of the PCON_CI and PCON_DI regressions on COD in Table 14 are consistent with the results of the OLS regression in Table 8. However, the PCON_CIDI regression shows a positive and significant relationship to COD.

## 5. Conclusions

This study examines the relationship between politically connected independent commissioners and politically connected independent directors to the cost of debt in all companies listed on the Indonesian stock exchange in the 2010–2017 period. Our OLS regression results show that both politically connected independent commissioners and politically connected independent directors negatively correlate with the cost of debt. We tested these results again using the Heckman 2SLS test, and the results were the same as our OLS regression. These results support our two hypotheses in this study.

This research has implications for the development of the literature related to corporate governance, political connections, and more specifically, to public companies that

independent commissioners and independent directors who are politically connected and must not only fulfill the requirements of establishing a public company, but also have an essential role on the board of commissioners and other boards of directors in the company in terms of making financial decisions.

This research certainly has limitations, which will be the basis for further research. First, the educational background of politically connected independent commissioners and politically connected independent directors needs to be analyzed further. However, both of these aspects have political connections. It is also necessary to have an education, especially in economics, management, and finance, to manage and monitor companies. Second, we re-examination considering family-owned and non-family-owned firms, which examines which one experiences a lower cost of debt. Third, it is necessary to examine the demographic variables of the board of commissioners or the board of directors (such as gender and origin) and the presence of women on the board of commissioners or directors in managing and making decisions within the company.

**Author Contributions:** Conceptualization, O.J. and I.H.; methodology, O.J. and M.A.; software, O.J.; validation, O.J., I.H. and M.N.; formal analysis, O.J.; investigation, I.H.; resources, I.H.; data curation, O.J.; writing—original draft preparation O.J.; writing—review and editing, O.J. and I.H.; visualization, O.J.; supervision, I.H., M.A. and M.N.; project administration, O.J.; funding acquisition, I.H. All authors have read and agreed to the published version of the manuscript.

**Funding:** This research received external funding from directorate of Research and Service, Ministry of Research and Technology/National Research and Innovation Agency of the Republic of Indonesia.

**Acknowledgments:** Acknowledgments to the Directorate of Research and Service, Ministry of Research and Technology/National Research and Innovation Agency of the Republic of Indonesia for the grant provided to finance this research.

**Conflicts of Interest:** The authors declare no conflict of interest.

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
