# Peer review of "Politically Connected Independent Commissioners and Independent Directors on the Cost of Debt"

_ijfs, doi:10.3390/ijfs10020041_

Round 1

Reviewer 1 Report

I would like to thank the authors for exerting significant effort to improve the paper. They have incorporated all comments raised and the manuscript has improved materially, relative to its previous version. Nevertheless, I urge authors to check the manuscript for any remaining syntax errors and the completeness of the reference list.

Author Response

Reviewer 1:

I would like to thank the authors for exerting significant effort to improve the paper. They have incorporated all comments raised and the manuscript has improved materially, relative to its previous version. Nevertheless, I urge authors to check the manuscript for any remaining syntax errors and the completeness of the reference list.

Reply: We have double checked the writing and send it for proofread to better present the writing. We also have checked the reference list with the body text.

Reviewer 2 Report

Dear Authors

Thank you for your revision of the paper.

Although the methodological choices and settings have significantly improved, the writing still requires significant editing.

Kind regards

Author Response

Reviewer 2:

Dear Authors. Thank you for your revision of the paper.

Although the methodological choices and settings have significantly improved, the writing still requires significant editing.

Kind regards

Reply: We have double checked the writing and send it for proofread to better present the writing.

Reviewer 3 Report

Summary

The paper explores the role of ‘independent commissioners’ and ‘directors’ of business corporations in Indonesia. The authors study whether this constellation impacts the cost of debt.

The paper has some methodological flaws.

Major Comments:

  • Lacks in scientific style: Abstract (shorter). The language. The literature is biased. It does not include seminal papers in the field;
  • Methodology I: The authors do not distinguish between different degrees of political connections. Does the annual report even provide sufficient information on that issue? What happens if the commissioner is independent, yet somebody in the board/firm is politically related to the government?
  • Methodology II: Table 8ff. R-square adj. is missing; no F-Tests for your regression models?, etc.

Data sample must be up-to-date for a journal publication in 2022. Your data ends in 2017!

Author Response

Reviewer 3:

Summary

The paper explores the role of ‘independent commissioners’ and ‘directors’ of business corporations in Indonesia. The authors study whether this constellation impacts the cost of debt.

The paper has some methodological flaws.

 Major Comments:

  • Lacks in scientific style: Abstract (shorter). The language. The literature is biased. It does not include seminal papers in the field;

Reply: Abstrak telah kami perbaiki

This study examines the relationship between politically connected independent commissioners and independent directors on the cost of debt in companies listed on the Indonesia Stock Exchange in the 2010-2017 period. The total sample is 327 companies with a total observation of 1,722 firm-year. This study uses the usual least squares regression model (OLS) and the Heckman 2SLS method. The results show that both politically connected independent commissioners and politically connected independent directors are negatively related to the cost of debt. We tested these results again using the Heckman 2SLS test, and the results remain consistent. This research contributes to the development of literature related to corporate governance, political connections, and public companies with politically connected independent commissioners and independent directors. This research is very important because it focuses on independent commissioners and independent directors, and have an important role in decision-making in companies.

  • Methodology I: The authors do not distinguish between different degrees of political connections. Does the annual report even provide sufficient information on that issue? What happens if the commissioner is independent, yet somebody in the board/firm is politically related to the government?

Reply: Thank you for your comments. I agree with your comments. However, given the limited information provided in the annual report, the best information that we can get is as presented in the paper. To highlight the contribution and to fill the gap on the literature, we choose to focus on the issue on independent commissioners.

  • Methodology II: Table 8ff. R-square adj. is missing; no F-Tests for your regression models?, etc.

Reply: We have added F-test to the tables.

  • Data sample must be up-to-date for a journal publication in 2022. Your data ends in 2017!

Reply: We have described the reason related to this issue as we present it on section 3. The year of data observation is limited to 2017 because, in 2018, IDX cancel the requirement of independent director position at the companies on the IDX. The detail of the policy changes is available at the Decree of the Directors of the Indonesia Stock Exchange Number: Kep-00183/BEI/12-2018.

This manuscript is a resubmission of an earlier submission. The following is a list of the peer review reports and author responses from that submission.

Round 1

Reviewer 1 Report

The paper under title “EXISTENCE OF INDEPENDENT COMMISSIONERS AND 2 INDEPENDENT BOARD OF DIRECTORS WHO ARE POLITI-3 CALLY CONNECTED WITH THE COST OF DEBT” deals with the examination of politically connected board and directors on the firms’ cost of debt within the Indonesian capital market. The paper is interesting in terms of the market under focus and the association of corporate governance, political affiliations and cost of debt. However, it has several weaknesses on theoretical background, empirical design and overall contribution to existing literature which make it unsuitable for publication in its current form. The paper requires several improvements in almost all parts, so I urge authors to consider the following comments for future potential endeavors.

  1. At first, the paper requires an extensive English-editing effort, preferably from a native speaker since there are numerous syntax and grammatical mistakes which make the paper confusing and difficult to read.
  2. The title needs a reconsideration since the word “existence” seems odd to be tested as cost of debt exists or not along with directors’ political connections. So, authors need to re-edit the title so as to be clear to the read the main goal of the paper.
  3. The introduction lacks a sound theoretical framework on political connections (PC) on governance and cost of debt. This fact also limits the contribution of the paper to existing literature, which is minimal or unclear. Also, authors need to explain why Indonesia is the focus of the study. Which specific features warrant its examination on these matters? All these issues need to be thoroughly explained in order to justify the need and contribution of the study.
  4. At the end of page 2, authors state that low interest rates firms must have power on the board of commissioners. I believe this is not the only factor determining low interest rates on borrowing. Authors need to efficiently explain why this is the case so as to justify H1 and H2. My previous comments on the importance of the theoretical framework echoes on this comment too. Both hypotheses are not sufficiently supported, especially the discussion on H2 does not even mention political connections. So, this part of the text needs more work.
  5. More details are also required regarding the sample selection procedure. How many unique firms are included in the sample?
  6. Also, the definition of politically connected directors needs further clarification. Why are military or police officers considered as politically connected? The same question can be posed for judges and prosecutors. This definition of PC is very vague, and this may be a reason why authors get the insignificant results at the end. PC must be connected to those people having direct connections to the political parties and governments. Also, one director is considered as PC if he/she has at least one such characteristic? Please explain.
  7. Table 1 must provide the exact definition of the financial variables (ratios etc.)
  8. The SIC codes on tables 2 and 3 must provide what exactly they mean. It is unclear as it is right now what SIC no. 2 is and so on.
  9. Is the COD variable correctly measured? On table 4 the mean value of COD is 5.086 so how the interest paid can be 5 times higher than the due capital of the loan? Looking also at the maximum value 1,3 thousand times more interest than the amount of debt?
  10. Authors need to provide an explanation as to why they reached the insignificant impact of PC on cost of debt. What does this actually mean about this market? Also, authors need to connect this finding with previous evidence on the field and explain the disagreement of their findings (if any) with other relevant studies. Are authors sure that the results are not spurious? Have they performed any additional sensitivity tests?

Reviewer 2 Report

Dear Authors

In its current state, your paper is hard to evaluate. Allthough the topic is interesting and there is still room to contribute to literature and policy, as is, your research is difficult to provide additional contributions. Namelly:

1 - You should consider a very thorough editing of your paper. Sentences come accross truncated. There are several typo and you lack significant flow and economic and logical reasoning in your papers.

2 - Your sections are incomplete and need to adhere more to satandard scientific writing: your introduction lacks a brief overview of results and contribution; your literature review is very very very poor (a couple of paragraphs) and your positioning in terms of results into the literature is absent.

3 - Your methodology is flawed and lacks transparency: examples include, your cost of debt is interest expense (?), rather than % cost; your first and second stage regressors are the same (?); etc

Kind regards

Reviewer 3 Report

Summary

The paper studies the role of ‘independent commissioners’ and ‘directors’ of privately organized business corporations listed at the stock exchange in Indonesia. Yet both actors are related (connected) to the government, parliament or leading political party. The authors study whether this constellation has an impact on the cost of borrowing/ cost of debt.

The topic is of interest to the journal, yet the paper needs major revision.

Major Comments:

  • The paper lacks partly in scientific style. For instance: Abstract must have ~200 words. The abstract should explain the field, the methodology and the major new results in shorter words. Moreover, the JEL-Codes after keywords are missing; The language must be active voice and present tense and not past tense and passive voice; the literature is biased an does not include seminal papers in the field published in leading international journals; the methodology has major flaws (see Nr.3);
  • Methodology I: The authors do not transparently define what they mean by ‘politically connected with the cost of debt’. This is a mystery even to experts in this literature. They measure it by the annual reports with a dummy-variable. Yet, the authors do not distinguish between different degrees of political connections! And does the annual report even provide sufficient information on that issue? What happens if the commissioner is independent, yet the CEO (or CIO, CFO) is politically related to the government? Do the results hold in general – you need a panel of different countries?
  • Methodology II: In case of multivariate OLS regression, the authors have to do first needed pre-testing, such as heteroscedasticity, autocorrelation, stationarity, omitted variables, etc. Even more important is to adjust the OLS regression due to heteroscedasticity and autocorrelation etc. And you cannot work with R-square only (adj. R-square,…); What about F-Tests? Please do and include the adjustments before re-submission. Without sufficient econometric care, I have to reject this paper.
  • Punchline: Wording used in this paper is inappropriate and does not follow the terminology in science. For instance: ‘independent commissioners’ are normally only used in the literature of public finance related to state authorities or state representatives.
  • Shorten the title and make it a spelling eye-catcher;

Minor Comments:

  • 38, 41, 44, 51, 53, etc. The paper has layout, language, citation, and spelling errors. Please read and check the paper carefully before re-submission.
  • List of reference is incomplete: seminal papers in this field are missing completely. There is no cited top-paper in this list. Please do not focus only on the Asian-related literature.
  • Literature list: Nr. 9, 10,… has no vol., no pages, etc.

Please check all references with 100 percent care before re-submission.
